# WIP: Smart Knowledge Capture for Industrial Applications⋆

Markus Stolze[1][0000−0003−3506−8024] and Matthias Gutknecht[2]

[1] OST, University of Applied Sciences of Eastern Switzerland, Oberseestrasse 10,
CH-8640 Rapperswil, Switzerland `markus.stolze@ost.ch`
[2] STAR AG, Wiesholz 35, CH-8262 Ramsen, Switzerland
`matthias.gutknecht@star-group.net`

**Abstract.** Knowledge Graphs are critical for delivering semantic technical documentation and product information supporting Industry 4.0 processes. Creation and editing of Knowledge Graphs is slow, costly, and requires trained authors. The capturing of knowledge required for maintenance and troubleshooting of products is time and cost intensive. The Smart Knowledge Capture tool suite will enable subject matter experts to update and maintain Knowledge Graphs directly, thus saving time and cost and making product knowledge management more agile.

**Keywords:** Knowledge Graph · ontology · semantics · collaboration · critiquing

## 1  Introduction

As illustrated in Figure 1 the history and evolution of technical communications can be explained as progression towards

- **increasing semantics of the data models** used and increasing granularity of those models.
- **increasing integration of data** from different sources (engineering, logistics, etc.).

Increasingly, information about technical products must be machine-interpretable to enable digital processes. Digitalization of industrial processes (commonly referred to as Industry 4.0) within and between enterprises has to rely on data and relations between data, which have:

- **unambiguous semantics**: digital processes interpret the meaning and content of information in a uniform manner;
- **semantic interoperability**: based on standards and ontologies, data and relations can be exchanged between products, systems, processes, stakeholders and organizations preserving its semantics.

---

⋆ Research supported by Innosuisse

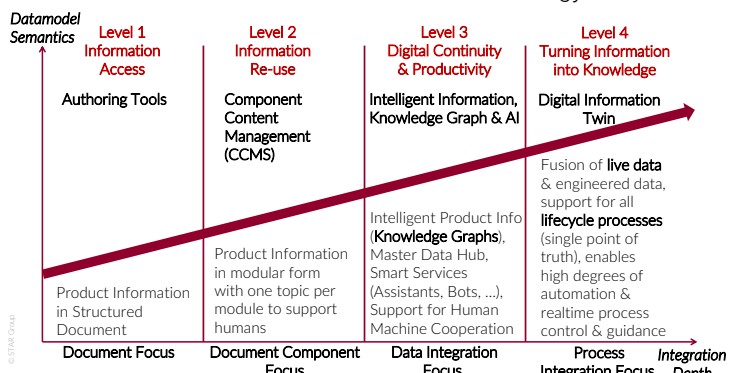

**Fig. 1.** Evolution of Technical Communication Technology

The German Standardization Roadmap Industry 4.0 [3] emphasizes the need for semantic unambiguity and interoperability. In order to fulfill these requirements, information about technical products has to be provided in a semantic structure. A frequently used data structure for semantically tagged data are Knowledge Graphs and Knowledge Graph-based ontologies [12]. To support the digitalization needs of the industry the trend is to replace or complement technical publications with so called digital information twins which provide:

- access to **product information for individual devices and systems (assets)**.
- access to **lifecycle information for individual assets** such as digital service history, as-maintained configuration information, results from condition monitoring and from big data analytics.
- **service oriented programmatic interface (API)** to use information and smart services provided by the digital information twin

Over the last twenty years, STAR AG has developed a comprehensive Knowledge Graph-based semantic meta model (see Figure 2 in sidebar) and associated tools to capture, manage and distribute semantic information, to support after-sales processes such as maintenance and troubleshooting. This model and the tools are successfully used by companies such as Daimler, Ferrari, Hilti, Liebherr, Vaillant, and Volvo Trucks. These companies also use the semantic information to enable new and smarter digital processes.

Currently capturing product knowledge and information in Knowledge Graphs, however, requires considerable training for technical authors, is rather slow and hence expensive, which is an obstacle to scale it up for all product information and for wider adoption in the market (including mid-sized companies). The challenge to be solved in the Smart Knowledge Capture project is to "shortcut" technical authors for all information "owned" by other stakeholders and subject

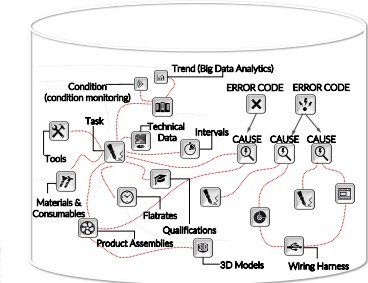

**Fig. 2.** Sample STAR Knowledge Graph

matter experts (SMEs) by providing them with a smart and easy tool to capture their knowledge in the Knowledge Graph (see Figure 3).

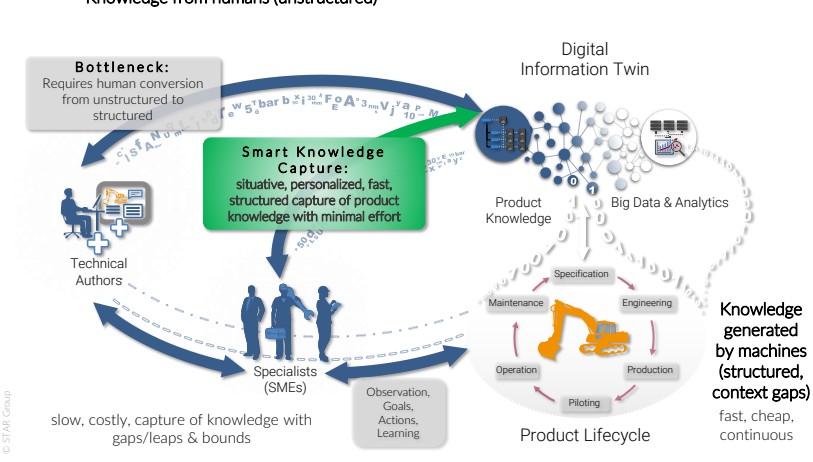

**Fig. 3.** Smart Knowledge Capture: Problem and Solution Sketch

## 2   Vision: Smart Knowledge Capture Tool-Suite

Our current vision for the Smart Knowledge Capture (SKC) tool-suite is shown in Figure 4. It consists of four main elements explained below: (1) the SKC Cloud Data Service, (2) the SKC Information Architects Suite, (3) the SKC Technical Author Suite, and (4) the SKC SME Points of Interaction.

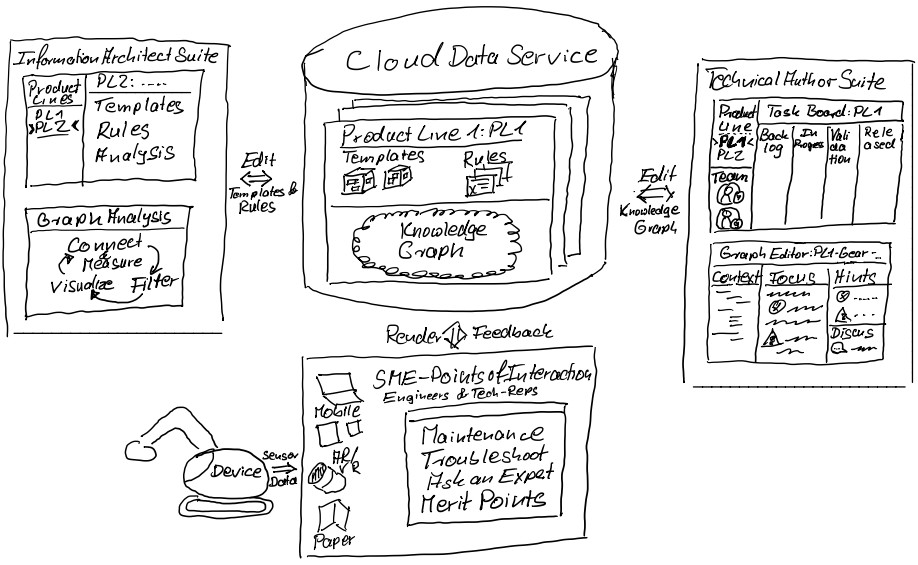

**Fig. 4.** Elements of the envisioned Smart Knowledge Capture Tool-Suite

The **SKC Cloud Data Service** provides the central data store for all the connected elements of the SKC Tool-Suite. In particular, it manages the associated Knowledge Graph for each line of industrial products, providing the semantically structured information about the associated products (technical specifications, component model, 3D model etc.) as well as information about maintenance and troubleshooting procedures and associated illustrations and view specifications. The SKC Cloud Data Service also stores the associated Templates and Validation Rules for each product line. These Templates and Rules are managed by the Information Architect responsible for the product line (see below) and are used and applied when Technical Authors extend or adapt the Knowledge Graph for a product line. *Templates* can determine their own applicability (matching rule) and provide a mechanism to quickly populate sections of the Knowledge Graph. They also provide means for contextual incremental specification where users are guided through a structured interview to complete the information required for filling in the template. *Validation Rules* are continuously checked during the editing of the Knowledge Graph. They provide warnings and hints to users about potential errors, inconsistencies or potential for optimization (e.g. reduction of redundancies).

The **SKC Information Architects Suite** is a collection of tools supporting the analysis and structuring of the Knowledge Graph for each product line. The Information Architect responsible for the product line uses the provided analysis tools to define a base structure for the Knowledge Graph as well as associated Templates and Validation Rules. Usually, the starting point will be a set of generic Templates and Rules which are then refined for the specific product

line. The Information Architect will monitor the development, extension and maintenance of the Knowledge Graph creating and adapting Templates and Rules as needed. To support this, task analysis and visualization tools will be provided.

*Research Challenges:* Currently, we are exploring and collecting possible candidate techniques for supporting the analysis of Knowledge Graphs so that Information Architects are supported in their task to identify needed Validation Rules and opportunities to provide Templates that boost the efficiency (and reduce error potential) of Technical Authors. Identified candidates include algorithmic techniques [4] and techniques developed for ontology engineering [9] [14] [2].

The **SKC Technical Author Suite** is a collection of tools supporting the Technical Authors in their collaborative task of creating, extending and maintaining the Knowledge Graph. The typical starting point of their work is the *Task-Board.* It provides an overview of the current editing tasks with multiple ways to filter the otherwise overwhelming number of editing tasks users are involved in. Examples of filters and sorting-rules include filtering by product, person, and type, and sorting by urgency or recency of comments. The Task-Board also helps Technical Authors to keep track of editing and validation tasks that they delegated to SMEs (see below) or colleagues. The *Graph Editor* supports the editing of the Knowledge Graph by providing three main panels: The Context panel, the Focus Panel and the Side Panel. The *Context Panel* shows the relevant contextual information for the current editing task. The *Focus Panel* supports the content editing. Here, applicable rules and opportunities for template application are indicated inline through visual annotations. The Side panel provides more detail about inline hints and collaborative annotations.

*Research Challenges:* An important research challenge for the SKC Technical Author Suite lies in the timely matching of rules and templates and the optimization of the attention-management. Here we plan to experiment with different search optimization techniques [8] [11] while adhering to guidelines for human-AI interaction [1] and visual critiquing [13]. We will also take into consideration findings from studies of general ontology editing tools [6] [16] [15] [10].

The **SME Points of Interaction** support Subject Matter Experts in their tasks. Development Engineers will be asked by Technical Authors to provide input regarding technical details and recommended maintenance and troubleshooting activities. They might also answer questions of Field Engineers and Technical Reps in the context of the Ask-an-Expert function. For contributions valued by the community "Merit Points" will be awarded. Field Engineers and Tech Reps will be the main users of the documentation provided, which is rendered in a device- and context-specific way from the information in the Knowledge Graph. When sensor information from the device is available, even more specific adaptation becomes possible.

*Research Challenges:* Challenges that we plan to address in the context of the SME Tools focus on the collaboration between Technical Author, Devel-

opment Engineer, Field Engineer and Tech Reps. We are aware that we must avoid well-known challenges of groupware [5]. A central question will be how to enable Development Engineers to contribute to the Knowledge Graph in a way that requires minimal training, but is still regarded as efficient, empowering, and motivating. We will track our progress by tracking the SUS [7] of our iterative prototypes. We will also investigate how to efficiently collect actionable feedback regarding maintenance and troubleshooting information from the Field Engineers and Tech Reps and how to best present this information to the Development Engineers and Technical Authors.

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
