# OpenReview forum: "WIP: Smart Knowledge Capture for Industrial Applications"
_eswc-conferences.org/ESWC/2021/Workshop/KGCW — Submitted to KGCW 2021_

### Official Review · ~Maxime_Lefrançois1 · 2021-04-12
**Workshop would be good for the authors, unsure the paper would be good for the audience**

**Rating:** 4
**Confidence:** 3

**Review:**

In this short vision paper, the authors depict a vision for the so-called "Smart Knowledge Capture tool-suite", which aims at helping subject matter experts (SMEs) structure their knowlege in a form that can be integrated in a Industry 4.0 knowledge graph. The ultimate goal is to limit the involvement of technical authors to convert unstructured data from the SMEs into structured data.

The paper is well positioned in the industry 4.0 roadmap, and well in the scope of the workshop: "End User Interfaces (UI) for (collaborative) editing and viewing for Knowledge Graphs building rules and management platforms in general"

The KGC workshop would definetely be a good venue for the authors to get feedback from the community on their ideas and the research challenges they identified. However given the very preliminary state of this reasearch, I am unsure the paper is worth publishing in its current form.

The tool-suite is in a very early design stage, and the paper only drafts a general architecture of the envisioned tool-suite, with four main components, and associated research challenges with a few spotted previous work, not really described, for each component. The reader is left with many questions unanswered about how the vision is meant to be realized. For example, I would like to know how the rules and templates look like, before heading to algorithmic techniques as identified in the research challenges

Minor issues:

- Fig 2 is too small
- Fig 4 is "cool", but not really readable.
- expand acronym SUS (System Usability Scale), which is not known by the KGC community
- references should be grouped, for example with command \cite{ref1,ref2} instead of \cite{ref1}, \cite{ref2}

---

### Official Review · ~Umutcan_Simsek1 · 2021-04-14
**good motivation, but more of a description of a GUI prototype than a vision**

**Rating:** 5
**Confidence:** 4

**Review:**

# content of the paper
The authors present their vision for a tool ecosystem that addresses knowledge acquisition task from domain experts. The major selling point of the proposed tool is that the knowledge acquisition (generation) task is partially shifted from technical authors to the domain experts which would remove some bottlenecks, particularly because of the communication between these two actors.

# strength
The paper is relevant to the workshop as it essentially provides UI for Knowledge Graph creation. It appears to also have a potential to gain some adoption among industrial practitioners as it is well aligned with the German national standardization roadmap for industry 4.0.

# main weakness
The tool is obviously at a very early stage, which would have been fine if the vision was described more clearly in an industrial and scientific framework. What I mean by this is actually already addressed in the paper but in a very limited way: the paragraphs at the end of each component descriptions called Research Challenges. The knowledge acquisition problem is perhaps the longest living problem in our field (also caused the AI winter) and there have been many attempts to solve it. Template and pattern based approaches are one of the most prominent ones (you can go back to work done with CyC, maybe even further).  I would perceive this paper more positively, if it identified shortcomings of the current paradigms regarding knowledge generation (acquisition) and what the drawn vision plans to offer to address these issues. In other words, these Research Challenges paragraphs could be extended and better structured.  Instead, the paper describes the vision based on a GUI prototype which makes it harder to demonstrate a benefit for a larger community, given that it is not even clear if the tool is going to be open-source.

# minor issues
- figures are unreadable. especially figure 2 and figure 4
- figure 3 suddenly talks about digital twins, which is not mentioned anywhere in the paper. how is the machine created data handled?
- are all data created manually? According to the figure 1 product data is structured. does that imply that some of the knowledge acquisition can be done via declerative mappings?
- how is the metadata evolving? only one organization is developing and maintaining it? Is it also a collaborative effort that is supported by the proposed tool suite?

---

### Official Review · ~Julián_Arenas-Guerrero1 · 2021-04-15
**Relevant topic, but introduces many concepts without explaining them enough**

**Rating:** 5
**Confidence:** 2

**Review:**

In this work the authors show the current state of a tool-suite for knowledge capture. They well motivate the need for such a tool (in Industry 4.0), and show that it has been successfully applied in industry. The authors outline the main challenges to be addressed to ease the work of people interacting with the tool suite.

In the Introduction the authors talk about what “digital information twin” provide. It is not clear to me what this is, is it the same than a digital twin? If yes, a description would be appreciated.

In the vision section, it is not clear enough the concept of “template” since the description is too general. What mechanism to populate the KG do you refer to?. When talking about “validation rules” what do you refer to with editing the KG? I assume you refer more to the ontology than to the KG (same with Graph Editor in section 2).

Sensor information for SME Points of Interaction is mentioned but not explained at all. The Research Challenges show lines in which further work is needed. It is difficult to see the alignment between problematic found in SKC elements and and the proposed challenges. More details and examples on SKC are needed.

In general, I would remove those specific terms that are not well explained, or provide more details/examples on them.

Figure 2 is too small. Figure 4 is nice but not easy readable. I also think that Figures 3 & 4 could benefit from a caption with a general description.

---

### Meta-Review · Program_Chairs · 2021-04-21

**Recommendation:** Reject
**Confidence:** 5

**Metareview:**

The reviewers agree that this is a relevant paper for the workshop but in its current state the paper is not ready for publication. The paper is framed as a position paper but according to the reviewers, the vision is not well explained while the paper is limited to the description of the implementation which does not have strong impact as it does not seem to significantly advance the state of the art.

---

### Decision · Program_Chairs · 2021-04-23

Reject